# Unfavorable Dynamics of Platelet Reactivity during Clopidogrel Treatment Predict Severe Course and Poor Clinical Outcome of Ischemic Stroke

**DOI:** 10.3390/brainsci11020257

**Published:** 2021-02-18

**Authors:** Adam Wiśniewski, Joanna Sikora, Aleksandra Karczmarska-Wódzka, Joanna Bugieda, Karolina Filipska, Robert Ślusarz

**Affiliations:** 1Department of Neurology, Faculty of Medicine, Nicolaus Copernicus University in Toruń, Collegium Medicum in Bydgoszcz, 85-094 Bydgoszcz, Poland; 2Experimental Biotechnology Research and Teaching Team, Department of Transplantology and General Surgery, Faculty of Medicine, Nicolaus Copernicus University in Toruń, Collegium Medicum in Bydgoszcz, 85-094 Bydgoszcz, Poland; joanna.sikora@cm.umk.pl (J.S.); akar@cm.umk.pl (A.K.-W.); joanna.bugieda@cm.umk.pl (J.B.); 3Department of Neurological and Neurosurgical Nursing, Faculty of Health Sciences, Nicolaus Copernicus University in Toruń, Collegium Medicum in Bydgoszcz, 85-821 Bydgoszcz, Poland; karolinafilipskakf@gmail.com (K.F.); robert_slu_cmumk@wp.pl (R.Ś.)

**Keywords:** platelet reactivity, clopidogrel, acute stroke, stroke severity, clinical outcome, prognosis, platelet function

## Abstract

Background: Previous studies have revealed that high platelet reactivity while on clopidogrel may affect the severe course and worse prognosis of ischemic stroke. However, the above findings were based on a single measurement of platelet function. We aimed to investigate whether the dynamics of platelet reactivity over time would more accurately determine its actual impact on clinical outcome. Methods: We enrolled 74 ischemic stroke subjects, taking a dose of 75 mg a day of clopidogrel to this prospective, single-center, and observational study. The determination of platelet function was based on the impedance aggregometry 6–12 h after the first dose of clopidogrel and 48 h later. We defined a favorable dynamics of platelet reactivity as a decrease in values at least equal to the median obtained in the entire study. The clinical condition was assessed by the National Institutes of Health Stroke Scale on the first, third, and ninetieth days and the functional status by modified Rankin Scale, respectively. Results: A favorable dynamics of platelet reactivity was associated with the mild clinical condition and favorable functional status, both early and late. Early neurological deterioration was related to unfavorable dynamics of platelet reactivity over time. In multivariate regression models, we found that unfavorable dynamics of platelet reactivity, alone and combined with a high baseline value of platelet reactivity, is an independent predictor of a severe clinical condition, the risk of deterioration, and poor early and late prognosis. Conclusion: We highlighted that dynamics of platelet reactivity over time predict the clinical course and prognosis of stroke better than a single value.

## 1. Introduction

The use of clopidogrel in the secondary prevention of ischemic stroke is steadily increasing. Previous studies demonstrated the efficacy and safety of clopidogrel administration, which enabled its wide application both in monotherapy and in combination with aspirin [1,2,3,4]. However, the combined use with aspirin in stroke is limited in time and strictly reserved for selected clinical cases [5]. Most often, this antiplatelet agent is included in the event of another recurrent cerebrovascular incident, in case of aspirin failure. High platelet reactivity on aspirin, also known as aspirin nonresponsiveness or aspirin resistance, is a common and widespread phenomenon that contributes greatly to the ineffectiveness of the applied antiplatelet therapy. The reduced platelet inhibitory effect, assessed in platelet function assays [6], may have important clinical implications. Many studies confirmed the significant negative impact of a low response to aspirin on stroke severity, poor clinical outcome, or higher risk of recurrent cerebrovascular events [7,8,9,10]. Our previous study demonstrated that the above findings are especially expressed in large-vessel disease, considered as an etiology of stroke [11].

The clinical significance of a similar phenomenon in relation to clopidogrel is also increasingly being raised. Considering that the mechanisms and background of the low responsiveness to clopidogrel are similar to those of aspirin, analogous and comparable clinical consequences should also be expected [12]. Other authors investigated the impact of clopidogrel resistance on the clinical course of stroke and reported that it is associated with worse clinical conditions and poor early and late prognosis [13,14,15,16,17]. However, the obtained results were based on a single-time-point determination of platelet function. In view of the emphasized high variability in platelet reactivity, a single measurement should be considered as a disadvantage and limitation [18,19]. Therefore, we hypothesize that an evaluation of the dynamics of platelet reactivity over time, estimated as a difference between two assessments, would more accurately and reliably determine the real importance of clopidogrel nonresponsiveness and better reflect its actual impact on the course of stroke.

The aim of the current study was to investigate the impact of dynamic changes in platelet reactivity over time during clopidogrel therapy on early and late clinical and functional conditions in stroke subjects.

## 2. Materials and Methods

### 2.1. Study Design and Participants

This study was conducted from November 2019 to November 2020 in a Stroke Center in the Department of Neurology at University Hospital No. 1 in Bydgoszcz, Poland. We included 74 ischemic stroke subjects who met both clinical and radiological criteria according to the updated stroke definition [20]. This research was single-center, observational, and prospective. Aspirin was administered to all subjects within the first 24 h and, then, from the second day after stroke symptoms, all subjects were treated with a 75 mg dose of clopidogrel. We enrolled participants with the following stroke etiology: large- vessel disease (at least 50% stenosis that covers the cerebral artery responsible for the symptoms of stroke) or small-vessel disease (typical morphological changes in the neuroimaging examination) [21]. We performed standardized, age-appropriate additional investigations to exclude other potential causes of ischemic stroke (including transthoracic or transesophageal echocardiography, 24–72 h Holter monitoring, genetic tests for thrombophilia, or laboratory findings specific for vasculitis).

We excluded stroke subjects who underwent specific therapy (intravenous thrombolysis and/or mechanical thrombectomy), who had a cardioembolic background of stroke (history of or recently detected atrial fibrillation, documented thrombus in heart ventricles), who were unable to sign the informed consent form (due to severe speech disorders or impaired consciousness), and who had contraindications to perform magnetic resonance imaging (e.g., pacemaker). The other exclusions included advanced neoplastic disease, documented history of severe bleeding (e.g., gastrointestinal), administration of antiplatelet agents before the current episode, previous stroke or transient ischemic attack in the last 3 years, and level of platelets below 100,000/µL. The general characteristics of stroke subjects are presented in Table 1.

### 2.2. Platelet Reactivity Research

Analysis of platelet function was performed by impedance aggregometry in the Laboratory of Experimental Biotechnology at Collegium Medicum in Bydgoszcz. Two blood samples, collected from the veins of the forearm, were used for platelet function testing. The first measurement was 6–12 h after the initial dose of clopidogrel. The second measurement of platelet aggregation was assessed 48 h later (±4 h). An adenosine diphosphate (ADP) test was applied in this study, where ADP was used as an activator of platelets in the Multiplate–Dynabyte multichannel platelet function analyzer (Roche Diagnostics, France). The addition of a platelet agonist to the solution activates them, forcing them toward the two electrodes, which is then perceived by the control system as a change in resistance (impedance). Next, automatic conversion of these signals is performed through graphical visualization as an area under the curve (AUC). The average for two electrode pairs was considered as the final result of the platelet function assessment. We adopted values over 46 AUC as the cutoff point, corresponding to high platelet reactivity on treatment, reflecting an insufficient inhibitory effect on platelets by clopidogrel. Similar values were used in other studies [22,23]. The processing steps of platelet function testing were consistent with those reported by other investigators [24]. The dynamics of platelet reactivity over time was assessed as the difference between values obtained in two measurements. A decrease in platelet reactivity equal to or higher than the median value obtained in this study (5AUC) was considered as a favorable change in platelet function. Any other observed dynamics were taken into account as an unfavorable change. Samples of favorable and unfavorable dynamics of platelet function are presented in Figure 1. Statistical calculations were used to highlight the significance of the dynamics of platelet reactivity over time.

### 2.3. Assessment of Clinical and Functional Condition and Adopted Definitions

The stroke severity was measured by the National Institutes of Health Stroke Scale (NIHSS) on the first, third, and ninetieth day after stroke onset. Disability was assessed by the modified Rankin Scale (mRS) on the first day, third day, and ninetieth day after stroke onset. We assumed the following definitions regarding clinical and functional condition:Early clinical condition—assessed by the NIHSS on the first and third days of stroke;Late clinical condition—assessed by the NIHSS on the 90th day of stroke;Early functional status—assessed by the mRS on the first and third days of stroke;Late functional status—assessed by the mRS on the 90th day of stroke;Severe stroke—NIHSS total score of 6 points or more;Mild stroke—NIHSS total score below 6 points;Favorable stroke—mRS total score below 3 points;Unfavorable stroke—mRS total score of 3 points and more;Deterioration in clinical condition—an increase of at least 1 point in the NIHSS score on the third day compared to the first;Deterioration in functional status—an increase of at least 1 point in the mRS score on the third day compared to the first.

### 2.4. Ethical Statement

The Bioethics Committee of the Nicolaus Copernicus University in Torun at Collegium Medicum of Ludwik Rydygier in Bydgoszcz (KB number 735/2019) approved the study protocol. All stroke subjects were able to read the study protocol and signed an informed consent form before enrollment to the study. The research was conducted according to the Declaration of Helsinki.

### 2.5. Statistical Evaluation Methods

The statistical calculations were performed using STATISTICA device, version 13.1 (Dell Company, Austin, TX, USA). The collected data are presented according to nonparametric characteristics, such as median and range. A Mann–Whitney U test and Spearman’s rank correlation test were assessed for evaluation of the dynamics of platelet reactivity over time and the relationships between continuous parameters, respectively. Univariate and multivariate logistic regression models were used for the estimation of predictive values of high platelet reactivity on clopidogrel. All variables that reached a *p*-value less than 0.05 or showed a trend (*p* <0.1) in univariate calculations were examined as the independent variables in multivariate models. A level of *p* <0.05 was considered statistically significant.

### 2.6. Definitions of Clopidogrel Nonresponsiveness

We introduced three definitions of clopidogrel nonresponsiveness used in logistic regression models. A high initial value of platelet reactivity (over 46 AUC) in the first measurement was the criterion for the first definition. Unfavorable dynamics of platelet reactivity over time (decrease lower than 5 AUC with every increase in value) was the basis for the second definition. The third definition included the simultaneous fulfillment of both of the above criteria.

## 3. Results

The median dynamics of platelet reactivity over time reached a decrease of 5 AUC, from 51 AUC (range 13–107 AUC) in the first measurement to 46 AUC (9–103 AUC) in the second measurement. The severity of stroke on admission (early clinical condition) assessed by total scores on the NIHSS significantly correlated with the value of platelet reactivity in the second measurement (*R* = 0.30, *p* = 0.0130) and did not correlate with the values reported in the first measurement (*R* = 0.20; *p* = 0.1016). No significant relationships were noted between the severity of stroke on the 90th day (late prognosis) and values of platelet reactivity in both assessments. Subjects with a severe early clinical condition (NIHSS 6 and over) were significantly more likely to achieve higher values of platelet reactivity compared to mild clinical condition, in both the first assessment and the second measurement (median 62 vs. 49 AUC, *p* = 0.0040; 66.5 vs. 41 AUC, *p* <0.0001, respectively). Similar dependencies were reported among subjects with severe late clinical condition (on the 90th day) (median 61 vs. 50 AUC, *p* = 0.04596; 68 vs. 44 AUC, *p* = 0.0002, respectively). A significant favorable dynamics of platelet reactivity over time was found in subjects with mild early severity of stroke (decrease from median 49 to 41 AUC; *p* = 0.0237) compared to unfavorable changes in platelet reactivity in subjects with severe early stroke (an increase from median 62 to 66.5 AUC; *p* = 0.5973) (Figure 2A,B). Similar dependencies were observed for late prognosis, when the severity of stroke on the 90th day was distinguished into mild and severe late clinical condition (decrease from median 50 to 44 AUC, *p* = 0.0243; an increase from median 61 to 68 AUC, *p* = 0.4013, respectively) (Figure 2C,D). Early deterioration in neurological status was associated with unfavorable dynamics of platelet reactivity (an increase from median 58 to 66 AUC, *p* = 0.2248), whereas stroke subjects without early deterioration were more likely to have significant favorable changes in platelet reactivity (decrease from median 50 to 44.5 AUC, *p* = 0.0296) (Figure 2E,F).

Disability on admission and on the 90th day (early and late functional status) assessed by total scores on the mRS significantly correlated with the values of platelet reactivity in the second measurement (*R* = 0.27, *p* = 0.0276; *R* = 0.30, *p* = 0.0131, respectively). No significant correlations were noted between disability and the first measurement of platelet function. Subjects with an unfavorable early functional status (mRS 3 and over) achieved significantly higher values of platelet reactivity compared to favorable status, but only in relation to the second measurement (median 55 vs. 41 AUC, *p* <0.0106), whereas subjects with unfavorable late functional status (on the 90th day) reached higher values of platelet reactivity, in both the first and the second measurement (median 59.5 vs. 49 AUC, *p* = 0.0235; 59.5 vs. 41 AUC, *p* = 0.0013, respectively). A significant favorable dynamics of platelet reactivity over time was found in subjects with early favorable disability (decrease from median 49 to 41 AUC, *p* = 0.0382) compared to unfavorable changes in platelet reactivity in subjects with early unfavorable disability (decrease from median 56 to 55 AUC, *p* = 0.8852) (Figure 3A,B). A similar relationship was noted for late functional outcome, where disability on the 90th day was distinguished into favorable and unfavorable functional status (decrease from median 49 to 41 AUC, *p* = 0.0412; identical median 59.5 vs. 59.5 AUC, *p* = 0.8802, respectively) (Figure 3C,D). Early deterioration in functional status was related to unfavorable dynamics of platelet reactivity (an increase from median 61 to 77 AUC, *p* = 0.1745), whereas stroke subjects without early deterioration exhibited a significant favorable dynamics of platelet reactivity (decrease from median 50 to 44.5 AUC, *p* = 0.0462) (Figure 3E,F).

In the univariate logistic regression, clopidogrel nonresponsiveness in the first definition did not affect the risk of occurrence of the selected characteristics of clinical and functional condition (Table 2). In contrast, clopidogrel resistance, according to the second and third definitions, significantly influenced the risk of occurrence of most or all characteristics, respectively. The predictors of selected characteristics of the clinical and functional condition were assessed by multivariate logistic regression. We developed two models adjusted for age, sex, etiology of stroke, common risk factors for vascular diseases, and clopidogrel nonresponsiveness according to the second (Model 1) and third (Model 2) definitions (Table 3). We excluded clopidogrel nonresponders in the first definition from further calculations due to insignificant relationships in univariate analysis. We revealed that clopidogrel nonresponsiveness was an independent predictor of a higher risk of clinical and functional deterioration, severe early and late clinical condition, and unfavorable early and late functional status.

## 4. Discussion

This is the first study to investigate the effect of dynamics of platelet reactivity over time on stroke severity and clinical outcome during clopidogrel therapy. We highlighted that changes in platelet reactivity better reflect their actual impact on the course and prognosis of stroke than a single baseline value. Therefore, it should be considered the main advantage and strength of the current research, and it represents a novel finding in this field. Previous studies only focused on a single measurement of platelet function. Due to the variability in platelet aggregation, a single assessment reduces the significance of the obtained results.

We reported that an unfavorable dynamics of platelet reactivity, defined as an increase or insufficient decrease over time, significantly affects poor early and late clinical and functional outcomes. Moreover, it is associated with a higher risk of early neurological deterioration. Furthermore, we emphasized that unfavorable dynamics of platelet reactivity remains an independent predictor of stroke severity and disability, exacerbation in clinical course, and unfavorable early and late prognosis. Our findings in the field of stroke are fully consistent with similar reports in the field of cardiology. We fully share the opinion of other authors that sequential measurements of platelet reactivity better reflect the essence of the phenomenon than a single result at one time point [25].

The baseline value of platelet aggregation did not correlate with stroke severity and disability, as assessed by total scores on the NIHSS and mRS, respectively. The second measurement turned out to be more reliable, as it was significantly more related to higher total scores in both scales. The division of strokes into mild and severe or favorable and unfavorable revealed higher values of platelet reactivity in the groups with poor prognosis, both early and late. Our findings are consistent with the results obtained by Jeon [14], who reported that stroke subjects with higher values of platelet reactivity exhibited higher total scores on the NIHSS on admission. In contrast, Yi et al. [17] showed that the value of platelet reactivity did affect the NIHSS total scores on admission. However, the overall impact of clopidogrel nonresponsiveness, defined as a higher baseline value of platelet reactivity, on the severity, disability, deterioration, and early or late prognosis among stroke subjects did not reach statistical significance in our study, as estimated in logistic regression. Our findings are inconsistent with other studies regarding this issue. Yi et al. [17] noted that clopidogrel nonresponders are more likely to have an unfavorable late outcome. However, they followed stroke subjects for up to 6 months. In our research, the follow-up period was shorter. Qui et al. [13] estimated that clopidogrel nonresponders exhibited poor late functional outcomes. However, they followed stroke subjects for up to 12 months and assumed different cutoffs for total scores on mRS (2 points or more) related to a poor outcome. Lee et al. [15] reported that clopidogrel nonresponsiveness is an independent predictor of early neurological deterioration. Their definition of deterioration was identical to that adopted in our study. Nevertheless, they investigated only stroke subjects with large-artery atherosclerosis and excluded other types of stroke. Similarly, Yi et al. [16] revealed that high platelet reactivity on clopidogrel is an independent predictor of early neurological deterioration. However, they adopted a different definition of deterioration (increase of at least 2 points on the NIHSS) and assessed the clinical progress within 10 days. Overall, in our opinion, there were a few additional issues that contributed to obtaining discrepancies: heterogeneous studied populations, a different methodology of platelet function assessment, small sample sizes, and minor statistical significance of the findings. The above inconsistencies confirmed the limited usefulness of a single measurement of platelet reactivity to properly assess the role of clopidogrel nonresponsiveness in the clinical course of a stroke.

In our study, only a combination of the initial value of platelet reactivity with the simultaneous assessment of platelet dynamics over time significantly influenced the clinical condition and prognosis of stroke subjects. Moreover, such a combination allowed achieving even greater statistical significance than in the case of assessing only the dynamics of platelet reactivity. Our findings prove that a high platelet reactivity on treatment is a factor that additionally enhances and strengthens the effect of the dynamics of platelet reactivity over time, as it does not have a significant impact itself. Furthermore, our study highlights the importance of proper platelet inhibition by clopidogrel in the course and prognosis of stroke. Only appropriate changes in platelet reactivity, in the form of a decrease in value over time, enable effective platelet inhibition, which is crucial to a favorable course of a stroke. An insufficient decrease or an increase in platelet reactivity values over time is associated with clopidogrel failure and significantly reduced effectiveness of the antiplatelet agent, which dramatically translates into a more severe course of a stroke, risk of deterioration, or unfavorable outcome.

Despite the study’s many strengths, we are aware of some limitations. Our findings are based on a small sample that needs to be confirmed in future research with a larger population. There is still no evidence for a standardized assessment of platelet function. Therefore, our results should be considered with caution, especially as they are based on a single device. In addition, due to the requirement of informed consent, our analysis was not conducted in the most severely affected subjects; thus, the entire spectrum of stroke is missing. We also take into account that a small percentage of subjects may have had a different stroke background that could not be detected during hospitalization.

## 5. Conclusions

In summary, we are the first to emphasize the significant impact of unfavorable dynamics of platelet reactivity during clopidogrel therapy on a severe clinical course and poor prognosis in ischemic stroke. We hypothesize that it is essential to maintain a beneficial inhibitory effect of clopidogrel over time in the acute phase of a stroke. Accordingly, our findings support the regular, individualized monitoring of platelet function in stroke to confirm the effectiveness of antiplatelet therapy and to predict clinical outcomes. Thus, our study sets the direction for further research into platelet function in this regard.

## Figures and Tables

**Figure 1 brainsci-11-00257-f001:**
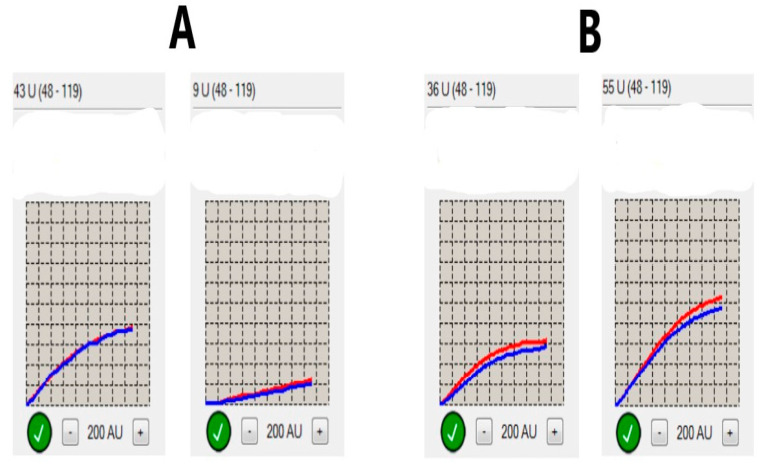
Dynamics of platelet reactivity over time. (**A**) Sample of favorable dynamics of platelet reactivity (defined as a decrease over time at least 5 U- area under the curve units). (Left) Normal baseline value of platelet reactivity (43 U). (Right) Decrease in value of platelet reactivity 48 h later to 9 U; the difference between both measurements is 34 U, which meets the adopted criteria of favorable dynamics. (**B**) Sample of unfavorable dynamics of platelet reactivity. (Left) Normal baseline value of platelet reactivity (36 U). (Right) Increase in value of platelet reactivity 48 h later to 55 U. The reported increase over time is considered as unfavorable change over time and reflects clopidogrel nonresponsiveness. The final value of platelet reactivity given as U is the average between values obtained in two pairs of electrodes (marked as red and blue lines).

**Figure 2 brainsci-11-00257-f002:**
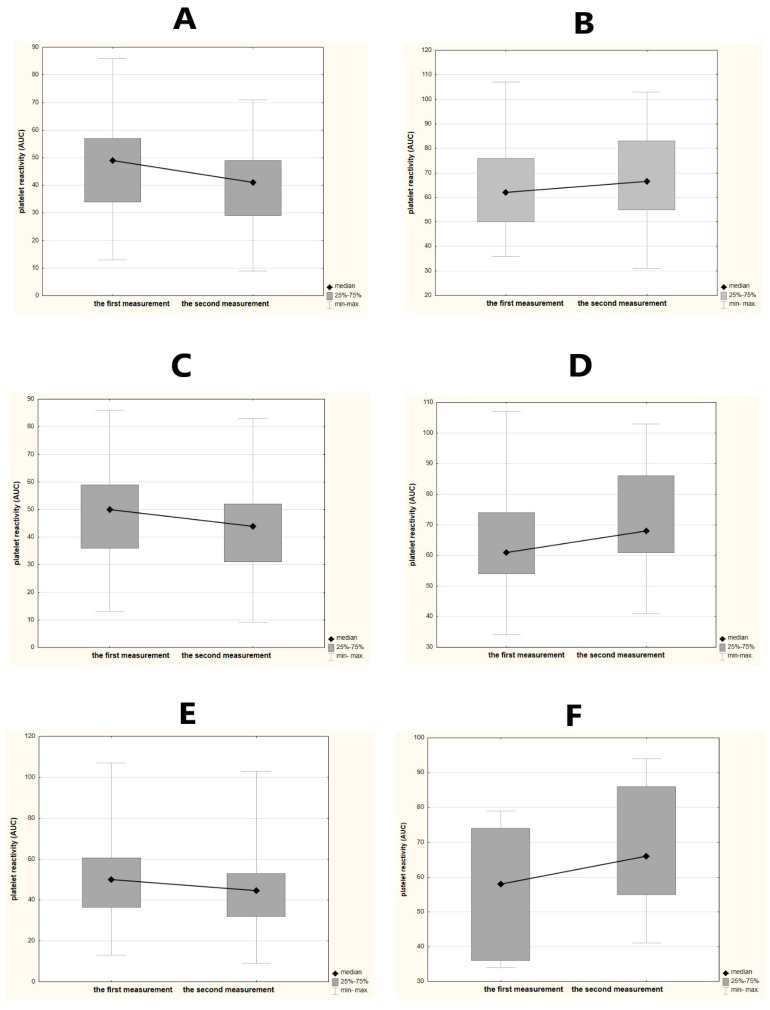
Dynamics of platelet reactivity over time in relation to the severity of stroke. The dynamics of platelet reactivity is the difference between two measurements of platelet reactivity over time (48 h). We defined a favorable change as a decrease over time at least 5 AUC (area under the curve units). Any increase over time or a decrease less than 5 AUC was defined as an unfavorable change. (**A**) A significant and favorable dynamics of platelet reactivity in stroke subjects with mild early stroke severity—the National Institutes of Health Stroke Scale (NIHSS) 0–5 on admission. (**B**) An unfavorable dynamics of platelet reactivity in stroke subjects with severe early clinical condition—NIHSS 6 and over on admission. (**C**) A significant and favorable dynamics of platelet reactivity in stroke subjects with mild late stroke severity—NIHSS 0–5 on 90th day. (**D**) An unfavorable dynamics of platelet reactivity in stroke subjects with a severe late clinical condition—NIHSS 6 and over on 90th day. (**E**) A significant and favorable dynamics of platelet reactivity in stroke subjects without early deterioration in clinical condition—no reported increase in total scores on the NIHSS between the third and first days. (**F**) An unfavorable dynamics of platelet reactivity in stroke subjects with early deterioration in clinical condition—increase in total scores on the NIHSS of at least 1 point between the third and first days.

**Figure 3 brainsci-11-00257-f003:**
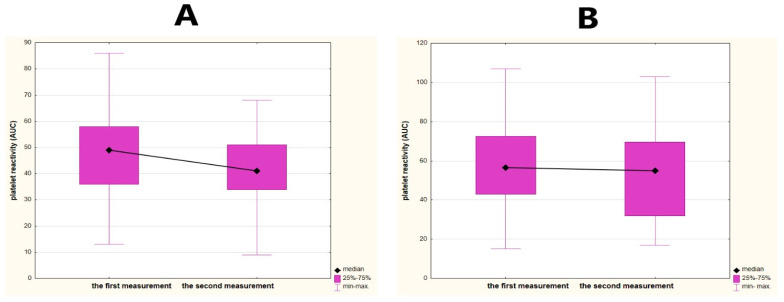
Dynamics of platelet reactivity over time in relation to the disability of stroke. The dynamics of platelet reactivity is the difference between two measurements of platelet reactivity over time (within 48 h). We defined a favorable change as a decrease over time at least 5 AUC (area under the curve units). Any increase over time or a decrease less than 5 AUC was defined as an unfavorable change. (**A**) A significant and favorable dynamics of platelet reactivity in stroke subjects with early favorable disability—the modified Rankin Scale (mRS) 0–2 on admission. (**B**) An unfavorable dynamics of platelet reactivity in stroke subjects with early unfavorable disability—mRS 3 and over on admission. (**C**) A significant and favorable dynamics of platelet reactivity in stroke subjects with late favorable functional outcome—mRS 0–2 on the 90th day. (**D**) An unfavorable dynamics of platelet reactivity in stroke subjects with unfavorable late functional outcome—mRS 3 and over on 90th day. (**E**) A significant and favorable dynamics of platelet reactivity in stroke subjects without early deterioration in functional status—no reported increase in total scores on the mRS between the third and first days. (**F**) An unfavorable dynamics of platelet reactivity in stroke subjects with early deterioration in functional status—increase in total scores on the mRS of at least 1 point between the third and first days.

**Table 1 brainsci-11-00257-t001:** The general characteristics of stroke subjects (*n* = 74).

Parameter	
Age, median (range)	67.5 (18–91)
Sex:	
Male, *N* (%)	36 (48.6%)
Female, *N* (%)	38 (51.4%)
Hypertension, *N* (%)	53 (71.6%)
Diabetes, *N* (%)	19 (25.7%)
Hyperlipidemia, *N* (%)	24 (32.4%)
Smoking, *N* (%)	22 (29.7%)
Obesity, *N* (%)	22 (29.7%)
Alcohol abuse, *N* (%)	6 (8.1%)
CRP (mg/L), median (range)	2.37 (0.21–148.96)
HbA1c (%), median (range)	5.8 (4.9–13.01)
D-Dimer (mg/mL), median (range)	451 (165–5926)
Fibrinogen (mg/dL), median (range)	335 (212–658)
Platelet count (thousands/µL), median (range)	271 (115–618)
NIHSS on admission, median (range)	3 (1–16)
NIHSS 3rd day, median (range)	2 (0–14)
NIHSS 90th day, median (range)	1 (0–14)
mRS on admission, median (range)	2 (0–5)
mRS 3rd day, median (range)	1 (0–5)
mRS 90th day, median (range)	0 (0–5)
Resistance to clopidogrel	
(initial value over 46 AUC), *N* (%)	47 (63.5%)
Etiology of stroke:	
Large-vessel disease, *N* (%)	18 (24.3%)
Small-vessel disease, *N* (%)	56 (75.7%)

CRP, C-reactive protein; HbA1c, glycated hemoglobin; mRS, modified Rankin Scale; NIHSS, the National Institutes of Health Stroke Scale; AUC, area under the curve; obesity, body mass index over 30; alcohol abuse, consuming at least 2 beers or 100 mL of vodka daily, most days a month for a period of 3 months.

**Table 2 brainsci-11-00257-t002:** The univariate logistic regression of the risk of selected characteristics of the clinical and functional condition among clopidogrel nonresponders vs. responders, depending on three different definitions of the nonresponsiveness.

	Definition 1		Definition 2		Definition 3	
	OR (95% CI)	*p*	OR (95% CI)	*p*	OR (95% CI)	*p*
Deterioration (NIHSS)	1.55 (0.28, 8.68)	0.6155	44.03 (2.37, 816.60)	0.0111 *	27.5 (4.20, 179.87)	0.0005 *
Deterioration (mRS)	7.48 (0.40, 141.28)	0.1795	28.6 (1.50, 544.87)	0.0257 *	115.0 (5.59, 2366.9)	0.0020 *
Severe early clinical condition	4.60 (0.94, 22.61)	0.0603	3.71 (1.09, 12.61)	0.0353 *	9.19 (2.11, 39.94)	0.0031 *
Severe late clinical condition	2.30 (0.44, 12.06)	0.3246	10.03 (1.87, 53.71)	0.0071 *	13.25 (2.67, 65.82)	0.0016 *
Unfavorable early functional status	1.75 (0.60, 5.09)	0.3054	1.85 (0.65, 5.27)	0.2528	5.49 (1.27, 23.80)	0.0229 *
Unfavorable late functional status	3.29 (0.83,12.97)	0.8972	3.76 (1.16, 12.13)	0.0267 *	7.05 (1.67, 29.68)	0.0077 *

* significant dependencies; mRS, modified Rankin Scale; NIHSS, the National Institutes of Health Stroke Scale; OR, odds ratio; CI, confidence interval.

**Table 3 brainsci-11-00257-t003:** Multivariate logistic regression analysis of predictors of selected characteristics of the clinical and functional condition in two models depending on the definition of the nonresponsiveness.

	Model 1		Model 2	
	Adjusted OR (95% CI)	*p*	Adjusted OR (95% CI)	*p*
(1) deterioration (NIHSS)				
Clopidogrel nonresponsiveness	46.82 (2.09, 1176.20)	0.0228 *	145.78 (6.38, 5944.85)	0.0062 *
Sex (male)	2.23 (0.18, 30.82)	0.4266	6.92 (0.18, 155.55)	0.2780
Age	0.96 (0.94, 1.04)	0.5689	0.97 (0.95, 1.02)	0.4523
Diabetes	0.03 (0.00, 1.01)	0.0638	0.05 (0.03, 3.02)	0.2635
Large vessel disease	0.14 (0.00, 1.86)	0.0812	36.12 (0.68, 1062.57	0.0725
Smoking	2.02 (0.15, 31.96)	0.6243	4.34 (0.18, 121.62)	0.4329
Hyperlipidemia	0.42 (0.05, 5.95)	0.5737	0.14 (0.00, 3.18)	0.2277
Hypertension	1.76 (0.11, 30.11)	0.6184	1.18 (0.02, 71.63)	0.9138
Obesity	2.34 (0.14, 58.89)	0.5738	2.45 (0.04, 124.26)	0.6866
(2) deterioration (mRS)				
Clopidogrel nonresponsiveness	27.22 (1.79, 389.24)	0.0264 *	196.45 (1.90, 209761.82)	0.0236 *
Sex (male)	0.55 (0.08, 5.73)	0.6334	4.23 (0.20, 90.14)	0.3891
Age	1.01 (0.97, 1.03)	0.6578	1.02 (0.97, 1.03)	0.6590
Diabetes	0.08 (0.00, 1.32)	0.0793	0.04 (0.00, 1.36)	0.1606
Large vessel disease	0.77 (0.08, 8.88)	0.8832	0.90 (0.03, 53.56)	0.9532
Smoking	0.38 (0.03, 4.22)	0.3591	2.21 (0.07, 120.86)	0.7170
Hyperlipidemia	0.48 (0.04, 2.99)	0.3992	2.18 (0.05, 124.87)	0.7890
Hypertension	0.96 (0.09, 10.56)	0.9285	0.48 (0.00, 40.58)	0.7890
Obesity	0.72 (0.08, 6.52)	0.8115	1.99 (0.09, 41.96)	0.7846
(3) severe early clinical condition				
Clopidogrel nonresponsiveness	33.72 (3.12, 428.55)	0.0072 *	36.21 (2.22, 456.03)	0.0067 *
Sex (male)	0.47 (0.09, 3.73)	0.5678	0.93 (0.16, 6.61)	0.9116
Age	0.97 (0.94, 1.04)	0.3965	0.98 (0.95, 1.04)	0.4935
Diabetes	150.99 (6.38,	0.0032 *	35.65 (2.59, 380.56)	0.0058 *
Large vessel disease	45.97 (4.45, 524.8)	0.0029 *	40.78 (4.74, 512.52)	0.0022 *
Smoking	6.17 (0.53, 45.34)	0.0968	6.14 (0.83, 44.90)	0.0913
Hyperlipidemia	3.78 (0.41, 27.71)	0.3630	1.78 (0.29, 8.43)	0.6997
Hypertension	0.14 (0.00, 1.48)	0.1625	0.15 (0.00, 2.01)	0.1828
Obesity	7.24 (0.92, 77.77)	0.1597	3.77 (0.45, 25.53)	0.2838
(4) severe late clinical condition				
Clopidogrel nonresponsiveness	62.55 (3.77, 1698.95)	0.0037 *	44.68 (3.78, 499.97)	0.0034 *
Sex (male)	0.95 (0.11, 7.99)	0.9348	2.13 (0.50, 15.50)	0.4829
Age	1.01 (0.99-1.02)	0.7845	1.01 (0.96, 1.05)	0.8245
Diabetes	70.58 (2.43, 1896.55)	0.0148 *	0.08 (0.00, 1.09)	0.0732
Large vessel disease	0.14 (0.01, 1.67)	0.1394	0.15 (0.02, 1.59)	0.0833
Smoking	3.70 (0.39, 44.79)	0.3591	4.01 (0.31, 50.69)	0.2454
Hyperlipidemia	2.77 (0.29, 21.56)	0.4863	0.69 (0.11, 6.86)	0.8918
Hypertension	0.48 (0.04, 6.63)	0.5507	0.53 (0.04, 5.60)	0.5794
Obesity	9.23 (0.55, 145.71)	0.1474	5.87 (0.32, 70.88)	0.2369
(5) unfavorable early functional status				
Clopidogrel nonresponsiveness	4.24 (1.01, 17.53)	0.0396 *	6.32 (1.45, 36.43)	0.0316 *
Sex (male)	0.76 (0.21, 2.48)	0.5876	0.88 (0.26, 2.99)	0.7294
Age	0.99 (0.96, 1.04)	0.6245	0.98 (0.94-1.05)	0.4598
Diabetes	21.39 (3.45, 129.78)	0.0026*	12.88 (2.77, 62.38)	0.0034*
Large vessel disease	7.45 (1.88, 35.08)	0.0145*	7.97 (1.91, 38.25)	0.0067*
Smoking	3.12 (0.60, 15.60)	0.1904	3.92 (0.62, 19.44)	0.1458
Hyperlipidemia	3.23 (0.80, 14.73)	0.0880	2.81 (0.60, 11.45)	0.1764
Hypertension	0.42 (0.08, 2.01)	0.2876	0.40 (0.09, 2.16)	0.2536
Obesity	3.22 (0.66, 17.13)	0.1561	3.12 (0.44, 16.74)	0.1829
(6) unfavorable late functional status				
Clopidogrel nonresponsiveness	11.94 (1.99, 66.07)	0.0121 *	9.33 (1.09, 53.68)	0.0108 *
Sex (male)	1.28 (0.24, 5.26)	0.7994	1.89 (0.47, 7.98)	0.4786
Age	1.02 (0.99, 1.04)	0.3168	1.01 (0.99, 1.03)	0.4685
Diabetes	20.12 (2.14, 177.18)	0.0126 *	9.32 (1.65, 49.66)	0.0149 *
Large vessel disease	8.96 (1.78, 49.72)	0.0135 *	7.59 (1.50, 39.36)	0.0174 *
Smoking	5.67 (0.79, 41.48)	0.0757	6.14 (0.89, 33.19)	0.0671
Hyperlipidemia	1.99 (0.46, 8.20)	0.4789	1.56 (0.38, 6.84)	0.7881
Hypertension	0.32 (0.06, 2.34)	0.3996	0.50 (0.07, 3.56)	0.5878
Obesity	2.52 (0.38, 14.58)	0.4235	1.67 (0.45, 8.80)	0.6963

* significant dependencies; mRS, modified Rankin Scale; NIHSS, the National Institutes of Health Stroke Scale; OR, odds ratio; CI, confidence interval.

## Data Availability

The data that support the findings of this study are available from the corresponding author upon request.

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
