# Peer review of "Unfavorable Dynamics of Platelet Reactivity during Clopidogrel Treatment Predict Severe Course and Poor Clinical Outcome of Ischemic Stroke"

_brainsci, 2021, doi:10.3390/brainsci11020257_

Round 1

Reviewer 1 Report

Title: Unfavorable dynamics of platelet reactivity during clopidogrel treatment predict severe course and poor clinical outcome of ischemic stroke

Manuscript ID: brainsci-1097470

This is an interesting manuscript about to investigate whether dynamics of platelet reactivity over time would more accurately determine its actual impact on clinical outcome.

The author suggests that dynamics of platelet reactivity over time better than a single value predict the clinical course and prognosis of stroke.

In spite of theses attractive reviews, some careful considerations should be made.

Major point
1. More concise but concentrated contents should be needed.

Too many items (NIHSS, mRS, early, late, deterioration…) were described redundantly.

Those dependent variables are important, but it is a little difficult and uncomfortable to read.

  1. Among a total 74 patients, the median age was 67.5 (18-91) years old.

To exclude patients with other determined or unknown etiology, which tests were performed?

For example, TTE, 24h-holter , TEE, cardiac CT, vasculitis laboratory findings, cancer marker…

Because the youngest patient was only 18 years old.

Then he or she may have other determined etiology in classic TOAST classification.

These young age stroke etiologies may affect the result.

  1. In addition, why some patient has elevated D-dimer level (5926 mg/mL)?

Did the author evaluate the septic condition including pneumonia or Urinary tract infection or malignancy?

  1. Are all included patients first-ever stroke patients?

If not, previous drug history may be important to interpretate the result.

  1. Why did the authors use the definition of Deterioration NIHSS as an increase of at least 1 point on the NIHSS on the third day compred to the first?

Because as the author mentioned, generally END (Early neurological deterioration) was defined as a worsening of 2 or more points on NIHSS or 4 or more points on NIHSS up to.

Since NIHSS scoring is a little subjective based on each physician, one point change can be ambiguous.

  1. In several clopidogrel function tests, VerifyNow or other method are commonly used.

Furthermore, the CYP2C19 detection panel is the most extensive on the market and covers seven known poor metabolizer alleles and one known rapid metabolizer allele.

Then what is the clinical impact of this result? Because as the author mentioned, serial measurement of of platelet reactivity over time would be more accurate. But it is more inconvenient and maybe more expensive.

  1. Please define the “obesity” and “Alcohol abuse” in table 1.
  2. In multivariate logistic regression, why did the authors adjust those variables (clopidogrel non-responders, sex, diabetes, LAA, smoking, HL, HTN, and obesity)?

In general, since age may affect the prognosis, it should be adjusted.

The effect of diabetes and obesity on clinical outcome may be intermingled.

Please explain the criteria for choosing the above variables.

  1. Figure legends.

The content of a caption should make it possible for your reader to interpret and understand the significance of a figure without reading the main text.

But in Figure 1-3, I cannot understand without reading the main text.

  1. The figure arrangement is poor. Reduce the picture size so that it fits within one page.

Reviewer 2 Report

Wisniewski et al. investigated how the dynamics of platelet reactivity overtime during clopidogrel treatment predict clinical outcome of ischemic stroke. By establishing multivariate regression models, the authors revealed that unfavorable dynamics platelet reactivity over time during clopidogrel therapy is an independent predictor of stroke severity and clinical outcomes. Overall, the results are convincing and of good quality, there are still a few things need to be improved.

  1. In figure 1, the authors show examples of favorable and unfavorable dynamics of platelet reactivity overall time. It is very difficult to understand how the units are calculated without any labeling on the graph. Although the authors said U is the unit number under the curve, labeling the graph will help the readers to understand how the favorable and unfavorable dynamics are. In addition, the standard of favorable that defines the favorableness by the median value 5AUC is lack of support. Is there any other evidence to support decrease of platelet reactivity by 5AUC is enough for improve clinical outcome?
  2. In figure 2 and 3, the authors showed the favorableness and significance of platelet reactivity in different clinical conditions. Two-way anova is a more clear way to compare the platelet reactivity between groups under the same parameter, such as A and B, C and D, E and F.

Reviewer 3 Report

Dear Authors very compliments for this paper. It is very interesting. Please the authors can describe if there are in diabetic vs non diabetic and by gender? However can indicate other therapy assumed at home ? Thank you. 

Round 2

Reviewer 1 Report

The author eagerly responded to the reviewer's suggestion.

But I still think that the multivariate analysis was not performed appropriately.

I agree the statistical analysis that all variables that reached a p-value less 0.05 or showed a trend (<0.1) in the univariate calculations were examined as the independent variables in multivariate models. 

Authors can select variables according to their research design, but those things must be consistent. If you selected obesity or diabetes as an independent predictors  because they are known to be related to the poor stroke outcome, age should be adjusted, too. 

 Because "age" is the most well-known prognostic factor in almost studies.

Reviewer 2 Report

The authors have provided thoughtful responses.

Author Response

Thank You for Your positive comments. We have made every effort to improve our paper according to the Reviewers suggestions.

Reviewer 3 Report

This paper is very interesting. Very compliments. The authors describe that Early neurological deterioration was related to unfavorable dynamics of platelet reactivity over time.  Please can indicate if unfavorable dynamics of platelet reactivity over time was related with worse outcome? Thank you 

Author Response

Thank You for Your positive comments. As we have shown in Table 3 clopidogrel nonresponsiveness is an independent predictor of severe early and late clinical condition, as well as unfavorable early and late functional status. This undoubtedly confirms  an association of clopidogrel high on treatment platelet reactivity with a worse outcome in stroke subjects.